# Metabolic Modulators in Cardiovascular Complications of Systemic Lupus Erythematosus

**DOI:** 10.3390/biomedicines11123142

**Published:** 2023-11-25

**Authors:** Sofía Miñano, Cristina González-Correa, Javier Moleón, Juan Duarte

**Affiliations:** 1Department of Pharmacology, School of Pharmacy and Center for Biomedical Research (CIBM), University of Granada, 18071 Granada, Spain; sofiaminano@correo.ugr.es (S.M.); cristinaglez@ugr.es (C.G.-C.); 2Instituto de Investigación Biosanitaria de Granada (ibs.GRANADA), 18012 Granada, Spain; 3Ciber de Enfermedades Cardiovasculares (CIBERCV), 28029 Madrid, Spain

**Keywords:** systemic lupus erythematosus, endothelial dysfunction, hypertension, immunometabolism

## Abstract

Systemic lupus erythematosus (SLE) is a multifactorial disorder with contributions from hormones, genetics, and the environment, predominantly affecting young women. Cardiovascular disease is the primary cause of mortality in SLE, and hypertension is more prevalent among SLE patients. The dysregulation of both innate and adaptive immune cells in SLE, along with their infiltration into kidney and vascular tissues, is a pivotal factor contributing to the cardiovascular complications associated with SLE. The activation, proliferation, and differentiation of CD4+ T cells are intricately governed by cellular metabolism. Numerous metabolic inhibitors have been identified to target critical nodes in T cell metabolism. This review explores the existing evidence and knowledge gaps concerning whether the beneficial effects of metabolic modulators on autoimmunity, hypertension, endothelial dysfunction, and renal injury in lupus result from the restoration of a balanced immune system. The inhibition of glycolysis, mitochondrial metabolism, or mTORC1 has been found to improve endothelial dysfunction and prevent the development of hypertension in mouse models of SLE. Nevertheless, limited information is available regarding the potential vasculo-protective effects of drugs that act on immunometabolism in SLE patients.

## 1. Introduction

Systemic lupus erythematosus (SLE) is a chronic autoimmune disease characterized by multisystemic inflammation and organ damage manifestations that affect the skin, joints, kidneys, heart, lungs, blood, and the central nervous system [1]. Autoantibodies targeting the cell nucleus are present in 99% of SLE patients, and autoantibodies specific to double-stranded DNA (anti-dsDNA) have been identified in more than 70% of patients [2]. Even though the presence of anti-dsDNA is predictive in 95% of SLE cases, a more extensive immune dysregulation is involved in the etiopathogenesis of SLE, although the exact cause of SLE remains unclear [3]. In a study by Wang et al., mitochondrial DNA (mtDNA) was detected in neutrophils with extracellular traps (NETs) [4]. The researchers observed elevated levels of anti-mtDNA antibodies in SLE patients compared to control subjects, which were significantly correlated with disease activity. Moreover, the presence of anti-mtDNA antibodies was disproportionately associated with lupus nephritis and showed a stronger correlation with the lupus nephritis activity index compared to anti-dsDNA levels [4].

SLE is a complex disorder influenced by a combination of hormonal, genetic, and environmental factors, primarily affecting young women (with a female-to-male ratio of 9 to 1) and typically manifesting during the reproductive years. The leading cause of mortality in SLE is cardiovascular disease (CVD), which can be attributed to a combination of risk factors such as hypertension, dyslipidemia, and a prothrombotic state. Increased atherosclerosis has already been demonstrated in SLE [5]. SLE has been linked to a proatherogenic lipid profile that includes elevated total cholesterol, triglycerides, low-density lipoprotein cholesterol (LDL-C), and high-density lipoprotein cholesterol (HDL-C) [6]. The most common abnormality seen in SLE patients is a decrease in HDL-C. The prevalence of dyslipidemia was 36% at the time of diagnosis and 60% at 3-year follow-up in a cohort of 918 SLE patients [7,8]. In this study, the prevalence of other traditional risk factors, such current smoking and diabetes, at enrollment was 13.7% and 3.4%, respectively, which was also increased through the follow-up [7,8]. Furthermore, metabolic syndrome is recognized as a proinflammatory state that may contribute to premature atherosclerosis and diabetes development in SLE patients, increasing their CVD risk. Metabolic syndrome is more common in SLE patients [9]. A recent systematic review and meta-analysis looked at the risk of CV events and CV risk factors in adult SLE patients. When compared to adults without SLE, the relative risk (RR) of hypertension was higher at 2.7, whereas the RRs of diabetes and metabolic syndrome were elevated but not statistically significant [10]. However, while traditional Framingham risk factors may contribute to CVD pathogenesis in SLE patients, they cannot fully explain the increased CVD risk in SLE patients [11]. Indeed, certain SLE-specific factors may play a role in CVD onset and progression in SLE patients. Additionally, another contributing factor to cardiovascular disease in SLE is the presence of elevated antiphospholipid antibodies, which can directly induce pro-inflammatory and prothrombotic effects on the endothelium and disrupt coagulation by inhibiting annexin A5, thereby negating its antithrombotic and protective effects [12].

Zhao M et al. reported that 71.9% of SLE patients had hypertension, and a significant portion of them, specifically 74.4%, were either undertreated or not treated at all [13]. Despite the high prevalence of hypertension among SLE patients, current hypertension management guidelines do not address the specific needs of individuals with autoimmune disorders like SLE. This has led healthcare providers to rely on recommendations designed for the general population, lacking data from comprehensive clinical trials within the SLE patient group [14]. Consequently, many patients with this autoimmune disease do not receive the appropriate antihypertensive medications they may require [15].

Nevertheless, limited research has been conducted to explore the underlying pathophysiological mechanisms responsible for hypertension in SLE [16]. It is possible that part of the challenge in controlling blood pressure in autoimmune disease patients is due to an incomplete understanding of the pathophysiological mechanisms in SLE. Emerging evidence suggests that T helper (Th)17 cells may play a role in the development of hypertension in SLE [17,18]. The activation, proliferation, and differentiation of CD4+ T cells are intricately regulated by cellular metabolism, and the use of metabolic modulators has shown potential for benefiting individuals with SLE-associated hypertension.

In this context, our focus is on the evidence supporting the notion that metabolic modulators, aimed at improving the impaired CD4+ T cell metabolism in SLE, could serve as a therapeutic alternative in managing hypertension in SLE patients.

## 2. Systemic Lupus Erythematosus and Arterial Hypertension

Hypertension stands as the leading risk factor for the progression of renal, vascular, and cardiac diseases in the general population [19], which is exacerbated by immune-mediated mechanisms in SLE patients. Multiple studies highlight the elevated prevalence of hypertension in women with SLE. Women aged 35 to 44 with SLE are 50 times more likely to experience a cardiac event, such as infarction or angina, than individuals of the same age without the condition [12,20]. Nonetheless, there is a scarcity of mechanistic studies on hypertension in SLE, and the precise underlying mechanisms remain unknown [21].

The renal pathological mechanisms related to autoimmune-induced hypertension remain incompletely understood. It is established that a loss of self-tolerance results in the production of autoantibodies. These autoantibodies form complexes, which deposit into tissues such as the kidneys, leading to the activation of other immune cells and the complement system. This, in turn, triggers the local secretion of inflammatory mediators, promoting chronic renal inflammation and oxidative stress [22]. These processes can potentially disrupt fluid and electrolyte balance in the kidneys or lead to renal vascular dysfunction, subsequently causing hypertension [23].

Various animal models of SLE have been employed to investigate the genetic and immunological mechanisms contributing to this autoimmune disorder. The NZBWF1 mouse model, utilized for over four decades, closely resembles human lupus nephritis. These mice exhibit features like immunocomplex deposition in glomeruli, high plasma levels of anti-dsDNA, albuminuria, and hypertension between 25–30 weeks of age [24]. In both female humans with SLE and the NZBWF1 mouse model, lupus nephritis precedes hypertension. This provides an opportunity to study the contributing factors to hypertension in the context of chronic renal inflammation [25]. However, like humans, it exists a disparity between nephritis and blood pressure in animal models of SLE. For instance, MRL/lpr, BXSB, and NZBWF1 mice develop glomerulonephritis, but only the NZBWF1 mouse model develops hypertension [26]. Hence, the NZBWF1 model is especially valuable for understanding the pathophysiology of hypertension in SLE.

Several factors contribute to hypertension in this model, including altered renal hemodynamics, endothelial dysfunction, changes in the inflammatory cytokine profile, oxidative stress, dysfunction in the adaptive immune system, and sex hormones [21]. However, it is important to note that hypertension in this lupus mouse model is not salt-sensitive [27]. This precludes the use of diuretics as a treatment option. Nevertheless, angiotensin-converting enzyme inhibitor drugs (ACEIs) like captopril and enalapril delay the onset of renal damage and reduce blood pressure, while cyclophosphamide treatment does not have effect in blood pressure [28,29]. Moreover, captopril reduced chronic renal lesions by reducing TGF-beta expression in the kidneys but had no effect on autoantibody production [30].

Recently, a murine model of lupus induced by topical administration of imiquimod, a toll-like receptor (TLR) 7 agonist, in the ears of BALB/c mice has been established [31]. These mice develop splenomegaly, autoantibody production, and glomerulonephritis with immunocomplex deposition within 4–6 weeks. Furthermore, they exhibit elevated levels of type I interferon, a result of TLR7 activation. It is worth noting that gain-of-function variants of TLR7 have been linked to lupus nephritis [32]. This model has also shown signs of endothelial dysfunction [33] and hypertension [18], indicating that TLR7 endosomal receptor activation, triggered by self-antigens (ds-DNA and ssRNA), may contribute to lupus-associated vasculopathy.

Additionally, we have observed an approximately 2.5-fold increase in TLR7 expression in the vascular wall (unpublished data) in hypertensive NZBWF1 mice. However, the relevance of this finding to the occurrence of endothelial dysfunction in this model remains unknown. In another murine model induced by pristane to mimic SLE, which is widely used to evaluate potential therapeutic agents resembling human idiopathic lupus syndrome [34], hypertension, chronic inflammation, and endothelial dysfunction were also observed [35].

There is limited evidence regarding the impact of first line SLE therapies on blood pressure. Corticosteroids, the most commonly used agents for all autoimmune diseases, have been associated with hypertension in SLE patients. Furthermore, women who had used prednisone for a longer period and had a higher cumulative dose of prednisone, as well as those who had previously had a coronary event, were more likely to have plaque [36]. On the other hand, mycophenolate mofetil (MMF) and hydroxychloroquine have demonstrated beneficial effects on blood pressure, independently of additional antihypertensive treatments and existing renal disease [37]. Several mechanisms by which hydroxychloroquine may protect against vascular injury have been proposed in mouse studies, including the reduction of vascular oxidative stress via antioxidant action in a mouse model of SLE [17,38] and the regulation of endothelial nitric oxide synthase in a mouse model of antiphospholipid syndrome [39]. Novel immunomodulatory drugs like belimumab and anifrolumab offer promising options for regulating immune system activation in SLE, but their effects on renal inflammation and hypertension in SLE have received limited investigation. Other new immunomodulatory drugs include JAK inhibitors (jakinibs), which inhibit the downstream cellular effects of various cytokines, including type I IFNs and cytokines involved in neutrophil development and function. Jakinib tofacitinib was shown to be effective in reversing lupus symptoms and vascular dysfunction in mice, as well as inhibiting netosis [40]. Short-term use of tofacitinib in SLE patients with mild–moderate disease activity was safe and improved arterial stiffness [41]; however, long-term clinical studies are needed to determine whether this treatment confers benefits for SLE patients addressing CV morbidity and mortality, especially given a reported link between thrombosis and JAK inhibition [42].

### 2.1. Role of Endothelial Function in SLE Hypertension

A substantial number of individuals with SLE exhibit signs of subclinical vascular disease that precede the onset of atherosclerosis. These subclinical changes include the development of endothelial dysfunction (with preserved vascular smooth muscle function) [43], arterial wall thickening, and coronary perfusion abnormalities [44]. Multiple studies emphasize the prominent impact on the endothelium in SLE.

A meta-analysis of 25 case-control studies encompassing 1313 SLE patients and 1012 healthy controls, utilizing a random effects model, revealed that SLE patients exhibited lower brachial artery endothelium-dependent flow-mediated dilation compared to healthy controls [45]. Notably, diabetes mellitus, renal disease, and diastolic hypertension are significant contributors to endothelial dysfunction in SLE patients [45]. However, whether endothelial dysfunction plays a causal role in SLE-associated hypertension or is merely an associated condition remains uncertain.

Possible mechanisms underlying endothelial dysfunction in SLE encompass netosis, endotoxemia, and an imbalance between pro-inflammatory and anti-inflammatory cytokines, leading to reduced nitric oxide (NO) bioavailability, endothelial leakage, and impaired endothelial repair (Figure 1). The decreased vasodilator response to acetylcholine in NZBWF1 mice is observed before the onset of proteinuria and elevated blood pressure, suggesting that these early vascular function changes may contribute to the development of hypertension in SLE [46]. The specific causes of endothelial dysfunction in the NZBWF1 model are not yet completely understood. Oxidative stress and proinflammatory cytokines have been proposed as potential factors contributing to hypertensive endothelial dysfunction in SLE.

Our research has indicated that endothelial nitric oxide synthase (eNOS) expression in the aorta of NZBWF1 mice remains unaltered [17]. However, we have observed a reduction in the phosphorylation of its activation site, Ser-1177, and no changes in the NO- cyclic guanosine monophosphate (cGMP) pathway signaling [47]. Additionally, we have identified an increased production of oxygen free radicals (ROS), mediated by the activation of the NADPH oxidase (NOX) system, which appears to be responsible not only for endothelial dysfunction but also for the development of hypertension. The administration of antioxidants (tempol + apocynin) restored endothelial function and reduced blood pressure in lupus mice [17]. Similarly, in pristane-induced SLE mice, endothelial dysfunction was associated with eNOS uncoupling and overexpression of NOX-1 [48].

The alterations in endothelial function observed in the NZBWF1 model may be linked to the vascular proinflammatory state triggered by circulating autoantibodies and proinflammatory cytokines (tumor necrosis factor (TNF)α, interferon (IFN)γ, interleukin (IL)-17, and IL21), or the infiltration of immune cells, particularly Th17 lymphocytes, into the vascular wall [47,49]. The proinflammatory cytokine IL-17 is known to induce endothelial dysfunction in the vasculature through Rho-kinase-mediated mechanisms [50], possibly due in part to increased ROS generation via NOX activation [51]. In contrast, IL-10, the primary cytokine released by Tregs, is known to attenuate NOX activity [52]. In SLE mice, elevated levels of plasma endotoxins were observed. Activation of TLR-4 in vessels with bacterial products such as lipopolysaccharide (LPS) can lead to increased NOX-dependent O_2_^−^ production and inflammation [53]. The endothelial dysfunction identified in human umbilical vein endothelial cells (HUVECs) acutely induced (within 24 h) by plasma from lupus patients with active nephritis is associated with ROS production generated sequentially by endoplasmic reticulum stress and NOX activation [54].

The innate immune system’s role in SLE pathogenesis is of special interest, with current research identifying neutrophils, neutrophil extracellular traps (NETs), and IFN signaling as disease progression and endothelial dysfunction drivers. By cleaving vascular endothelial cadherin with neutrophil elastase present within NETs, NETs can promote vascular leakage and endothelial-to-mesenchymal transition, increasing beta-catenin signaling, which may have profibrotic effects in SLE patients [55]. Furthermore, endothelial progenitor cell loss and dysfunction have been linked to elevated serum levels of type I IFNs in SLE patients [56] and in mouse models of lupus [57,58], suggesting a role for SLE-specific immune dysregulation in contributing to an imbalance of increased endothelial damage and reduced vascular repair/angiogenesis. In addition, IFN-α has been shown to alter NO signaling via transcriptional control of eNOS expression, resulting in decreased NO generation in insulin-stimulated HUVECs, which contribute to endothelial dysfunction [59].

Large artery stiffness is caused by the deterioration of the anatomical features of the arterial wall, which results in a decrease in the elastin-to-collagen ratio. Vascular stiffness is an outcome of hypertension rather than its cause. When compared to controls, SLE patients had higher arterial stiffness [60,61]. Arterial stiffness, as well as associated features of reflected pulse waves in the artery, may contribute to systolic blood pressure and pulse pressure rises. Endothelial dysfunction and collagen deposition in the extracellular matrix are both important factors in arterial stiffness. Moreover, in young patients with SLE, there was a significant positive correlation of arterial stiffness with triglyceride-rich lipoproteins, although the cause–effect relationship of this link was not established [62].

### 2.2. Immune System and Hypertension in SLE

It is now commonly accepted that aberrant immune system activation is linked to the development of hypertension, both in humans and experimental models. In fact, there have been numerous genetic polymorphisms associated with hypertension described that exert effects on the immune response [63]. Non-specific immunosuppressive therapies, such as cyclophosphamide, have been proven in experimental models of spontaneous non-autoimmune hypertension to not only prevent the development of hypertension but also to lower blood pressure in animals with established hypertension [64]. Similarly, MMF, an immunosuppressive therapy frequently used in SLE patients to suppress T and B lymphocyte proliferation, has been effective in attenuating the development of hypertension in various experimental models and in humans [65,66,67]. Several studies have found a link between circulating autoantibodies, which are seen in systemic autoimmune diseases, and essential hypertension in humans.

SLE is a chronic autoimmune disorder characterized by the hyperreactivity of B and T lymphocytes due to a loss of tolerance to self-antigens, resulting in the production of pathogenic autoantibodies, particularly against nuclear components. However, it remains unclear whether the autoantibodies produced in SLE contribute mechanistically to the development of hypertension in these patients. The immunization process leading to the abnormal production of autoantibodies in SLE is partly attributed to the delayed clearance of apoptotic cells. Our research has demonstrated that the activation of peroxisome proliferator-activated receptors β/δ (PPARβ/δ) plays a crucial role in the proper clearance of deceased cells by macrophages in the NZBWF1 model, which has been associated with a reduction in elevated blood pressure [47].

Animal models of SLE are critical in understanding the relationship between autoantibodies and hypertension. As previously stated, the NZBWF1 mouse model closely resembles key characteristics of clinical SLE, including autoantibody generation, immune-complex-mediated kidney damage, and hypertension. Recent studies have shown that long-term B cell depletion using anti-CD20 successfully reduced autoantibody synthesis and prevented the onset of hypertension in SLE mice [68]. Furthermore, continuous administration of the immunosuppressive medication MMF selectively lowered B cells while preventing the development of hypertension [69]. These data unequivocally indicate a relationship between B cells, autoantibodies, and the onset of hypertension. However, it is important to highlight that these treatments were only beneficial when started before the commencement of the disease. Similarly, treatment efforts targeting B cells in humans, such as anti-CD20 (rituximab), have had minimal effectiveness in large controlled clinical studies [70]. It has been suggested that the low efficacy is due in part to the persistence of plasma cells, which are not the target of B cell treatments.

Plasma cells take up residence in the bone marrow and spleen for extended periods, ranging from months to years, and are primarily responsible for producing serum immunoglobulins, including SLE autoantibodies. The depletion of plasma cells using the proteasome inhibitor bortezomib has been demonstrated to diminish autoantibody production and alleviate hypertension in female NZBWF1 mice. Collectively, these findings indicate that the mechanistic link between autoantibody production and autoimmune-associated hypertension involves plasma cells [71].

Autoantibody production and the clinical severity of lupus in NZBWF1 mice prone to the disease are contingent on the assistance of CD4+ T cells [72]. Activated B cells migrate to areas rich in T cells within secondary lymphoid organs, where they recruit antigen-bound T cells. Subsequently, B cell differentiation occurs, either extrafollicularly into plasmablasts, with limited antibody production capacity, or by transitioning along the follicular pathway, forming germinal centers (GCs). Within GCs, follicular T-helper (Tfh) cells facilitate somatic hypermutation and isotype switching of B cells, culminating in the generation of long-lived plasma cells capable of producing high-affinity antibodies [73,74]. In a murine model of SLE, a connection between increased Tfh cell numbers and the development of autoimmunity has been established [75,76], suggesting that anomalies in the positive selection of Tfh cells could lead to systemic autoimmunity. Tregs also play a role in regulating the production of lupus-associated antibodies, including anti-dsDNA [77]. A substantial inverse correlation has been observed between Tregs and anti-dsDNA levels [78]. Although the precise role of Tregs in the pathogenesis of SLE remains to be fully elucidated, expanding Tregs could pose a potential challenge in the treatment of SLE, especially in terms of inducing B cell tolerance. Additionally, it has been postulated that Th17 cells, a subset of T cells, might be responsible for the abnormal selection of autoreactive B cells in GCs and the humoral response in vivo [79]. Collectively, inhibiting the differentiation of Tfh cells into Th17 cells and B cells into long-lived plasma cells, alongside expanding Tregs, could emerge as novel treatment options for SLE, centered on diminishing autoantibody production. Furthermore, the expansion of Treg cells through low-dose IL-2 has been shown to alleviate hypertension in NZBWF1 mice [80], while the neutralization of IL-17, the primary proinflammatory cytokine produced by Th17 cells, has been found to ameliorate endothelial dysfunction and reduce high blood pressure in SLE mice induced by TLR7 activation [18].

A vital clinical objective for patients with autoimmune diseases is the induction of tolerance. To attain this objective, a monoclonal antibody to CD3+ (anti-CD3), a subunit of the T cell co-receptor complex expressed on the surface of all T cells, has been utilized in both preclinical and clinical studies. It induces peripheral tolerance by expanding Treg cells and promoting the elimination of apoptotic bodies. Treatment of female NZBWF1 mice with anti-CD3 has reduced T and B cell hyperactivity, circulating anti-dsDNA levels, and the development of hypertension, irrespective of changes in renal injury. These findings suggest that anti-CD3 therapy, which mitigates immune system hyperactivity during autoimmune disease, may confer clinical advantages in attenuating cardiovascular risk factors, such as hypertension [81].

Conventional dendritic cells (DCs) are antigen-presenting cells that regulate the function of immature CD4+ T cells through cytokine production. The depletion of DCs in lupus MRL/lpr mice has been found to reduce autoimmune pathology and levels of autoantibodies. Surprisingly, this depletion had little impact on spontaneous CD4+ T cell activation, indicating that B cells, rather than DCs, are responsible for activating autoreactive CD4+ T cells. B cells activate CD4+ T cells differently than DCs. In general, B cells may play a more crucial role in CD4+ T cell activation in lupus, at least partly due to their chronic activation via the self-antigen-induced stimulation of endosomal TLR7 and TLR9 receptors mediated by the adaptor myeloid differentiation primary response protein 88 (MyD88). It has been noted that in vivo, B cells from lupus mice promote enhanced differentiation of CD4+ T cells into Th1 and Tfh cells while limiting the expansion of Treg cells to a greater extent than B cells from non-lupus mice [82].

In both SLE patients and SLE murine models, there was a significant enrichment of isolevuglandin-adducted proteins (isoLG adducts) in monocytes and DCs. Treatment of SLE-prone animals with the selective isoLG scavenger 2-hydroxybenzylamine (2-HOBA) improved several autoimmune measures, including plasma cell growth, circulating IgG levels, and anti-dsDNA antibody titers. Furthermore, 2-HOBA reduced blood pressure, reduced kidney damage, and decreased inflammatory gene expression, notably in C1q-expressing DCs. As a result, isoLG adducts play an important role in the development and maintenance of systemic autoimmunity and hypertension in SLE [83].

Neutrophils from MRL/lpr mice produce NETs faster than control mice. These MRL/lpr mice also generate autoantibodies against NETs and show signs of endothelial dysfunction. Inhibition of peptidylarginine deiminase (PAD) reduces NET formation and safeguards against lupus-related vascular damage in the New Zealand Mixed model of lupus [84] and MRL/lpr mice [85].

Several recent studies suggest that the gut microbiota may contribute to the onset of symptoms and the progression of autoimmune disease in both human and mouse models of SLE [86,87,88,89,90,91]. Notably, we have reported that the gut microbiota plays a role in the development of hypertension in both female NZBWF1 mice [92] and SLE mice induced by TLR7 activation [93]. In both murine models, the vascular changes induced by hypertensive SLE microbiota were linked to Th17 infiltration in the vasculature. Strategies aimed at modifying the gut microbiota composition in SLE, through the chronic consumption of probiotics, prebiotic fibers, or postbiotics, resulted in the restoration of Th17 polarization in gut secondary lymphoid organs. This, in turn, reduced Th17 infiltration in the vascular wall, improved vascular function, and lowered high blood pressure [49,94,95,96]. In summary, these findings suggest that the dysregulation of innate and adaptive immune cells in SLE and their infiltration into kidney and vascular tissues are key events associated with cardiovascular complications in SLE.

## 3. The Role of Cellular Metabolism in the Pathogenesis of SLE

As previously mentioned, autoreactive CD4+ T cells play a crucial role in SLE. Consequently, both SLE patients and mice prone to lupus exhibit elevated numbers of activated CD4+ T cells [97], and increased populations of Th1 and Th17 lymphocytes [98]. Furthermore, lupus-prone mice display signaling deficiencies in CD4+ T cells [99]. Mice lacking IFN-γ or IL-17 are protected against autoantibody production and the development of glomerulonephritis. Therefore, reducing Th1 and Th17 cells presents a promising approach for SLE treatment, involving the reduction of CD4+ T cell activation, proliferation, and differentiation upon antigen exposure.

These processes in CD4+ T cells are intricately regulated by cellular metabolism (Figure 2). Quiescent T cells have a low energy demand and can support oxidative metabolism with glucose, lipids, and amino acids. T cell receptor activation increases glycolysis and mitochondrial metabolism. Overexpression of the glucose transporter Glut1 in mice resulted in CD4+ T cell hyperactivation, hypergammaglobulinemia, and immunological complex-mediated nephritis, demonstrating the importance of glucose metabolism in autoimmunity [100]. Cellular metabolism also governs the differentiation of effector T cells and the formation of memory cells. Th1, Th2, and Th17 cells primarily rely on glycolysis, while Treg cells and long-lived memory T cells exhibit higher rates of lipid oxidation [101]. As a result, CD4+ T lymphocytes rely on glycolysis to perform inflammatory effector tasks. However, the role of these mechanisms in lupus is unknown. When lupus-prone B6.Sle1.Sle2.Sle3 (TC) mice are compared to non-autoimmune controls, both glycolysis and mitochondrial oxidative metabolism are increased in CD4+ T cells.

The triple congenic TC mouse model (*Sle1*, *Sle2*, and *Sle3*) carries three NZM2410-derived lupus susceptibility loci (*Sle1*, *Sle2*, and *Sle3*) on a non-autoimmune C57BL/6 (B6) strain. TC mice spontaneously develop symptoms similar to SLE patients, such as anti-dsDNA immunoglobulin G production and a high rate of fatal immune-complex-mediated glomerulonephritis. The *Sle1c2* susceptibility locus corresponds to *Esrrg* gene expression, which contributes to CD4+ T cell activation and enhanced IFN-γ secretion. *Esrrg* controls cell metabolism by increasing mitochondrial oxidative phosphorylation [102].

Inhibition of the glycolytic pathway promotes differentiation into Treg cells, which are deficient in SLE. Treg cells rely on exogenous fatty acids and mitochondrial metabolism during differentiation [101].

Glutamine is an essential substrate for lymphocyte functions, playing a vital role in differentiating CD4+ T lymphocytes into inflammatory cell types, such as Th1 and Th17 [103]. Glutaminase 1 (Gls1), the first enzyme in glutaminolysis, converts glutamine to glutamate. Kono et al. demonstrated that Gls1 inhibition with bis-2-(5-phenylacetamido-1,3,4-thiadiazol-2-yl)ethyl sulfide (BPTES) reduced Th17 differentiation and ameliorated lupus-like disease in MRL/lpr mice [104]. Additionally, enhanced mitochondrial functions, such as oxidative phosphorylation through glutaminolysis in B cells, induce plasmablast differentiation, correlating with disease activity scores in SLE patients. BPTES reduced oxidative phosphorylation, ROS production, and plasmablast differentiation, suggesting its potential as a therapeutic agent for SLE [105].

The understanding of how cellular metabolism influences immune cell activity has advanced dramatically in recent years, with T cells leading the way among other immune cell types [106]. Quiescent T cells produce ATP via oxidizing glucose, fatty acids, or glutamine. Several metabolic inhibitors impair important nodes of T cell metabolism (Figure 2). T cell metabolism has been recommended as a target for immunotherapy due to the dramatic variations in metabolic requirements and the crucial function of effector T cells in autoimmune disorders [107].

## 4. Metabolic Modulators and SLE

Multiple defects in immune metabolism have been observed in lupus patients and murine models of the disease [108]. Dysfunctional CD4+ T cell metabolism has been identified, making it a potential therapeutic target in both murine and human SLE. However, aside from CD4+ T cells, metabolic modulators can also directly impact other immune cell types in vivo. Glucose metabolism is crucial for B cell functions [109]. Both glycolysis and mitochondrial metabolism also play a vital role in the activation and maturation of DCs [110,111], which can indirectly affect T cells.

Metformin, a mitochondrial chain complex I inhibitor, and 2-deoxy-D-glucose (2DG), a glucose metabolism inhibitor, both inhibited IFN-γ production in vitro, though at different phases of activation. In vivo, the treatment of TC mice and other lupus models, including NZBWF1, with metformin and 2DG restored T cell metabolism and disease biomarkers, albeit their impact on the development of vascular dysfunction and hypertension was not tested. Furthermore, SLE patients’ CD4+ T cells had enhanced glycolysis and mitochondrial metabolism, which was associated with their activation state. Metformin considerably reduced their excessive IFN-γ production in vitro. Metformin also inhibited the type 1 IFN response in human CD4+ T lymphocytes via interfering with mitochondrial respiration [112]. These findings suggest that normalizing T cell metabolism through the dual inhibition of glycolysis and mitochondrial metabolism is a promising therapeutic target for SLE [113]. Moreover, treatment with the lactate production inhibitor, dichloroacetate, did not effectively prevent or reverse autoimmune pathology [114].

Metformin inhibited systemic autoimmunity in Roquinsan/san mice, a novel murine SLE model, by suppressing the differentiation of marginal zone B (MZB) cells and B lymphocytes into plasma cells, leading to a significant reduction in GC formation. Concerning T cells, metformin treatment resulted in a significant decrease in Tfh and Th17 cell populations, while increasing the Treg population. Metformin treatment also elevated AMP-activated protein kinase (AMPK) activity in splenic CD19+ B cells and CD4+ T cells in Roquinsan/san mice, while attenuating the expression levels of their downstream mammalian target of rapamycin (mTOR)/signal transducer and activator of transcription protein 3 (STAT3) signals. The mTOR signaling operates through two complexes: mTOR complex (mTORC)1 and mTORC2. mTORC1 is crucial for Th17 differentiation and inhibits Treg differentiation by suppressing Foxp3 expression. In SLE T cells, mTORC1 activity is increased, whereas mTORC2 activity is decreased. These findings suggest that an AMPK induction strategy could offer a novel therapeutic approach for lupus nephritis [115]. Recent research has demonstrated that metformin enhances the immunomodulatory potential of adipose tissue-derived mesenchymal stem cells, thus increasing their protective effects in lupus MRL/lpr mice [116]. Additionally, metformin alleviated kidney injury in lupus nephritis by suppressing renal necroptosis and inflammation through the AMPK/STAT3 pathway [117]. Notably, it has been observed that the protective effect of glycolysis inhibition in lupus is transferable through the gut microbiota, directly linking alterations in immunometabolism to gut dysbiosis in the hosts [118]. In a clinical trial, the metformin add-on group exhibited a 51% reduction in the frequency of flares in lupus patients compared to the conventional treatment group (comprising corticosteroids and immunosuppressive agents). Metformin demonstrated steroid-sparing effects, although no significant differences were observed between the two groups regarding overall prednisone exposure. Furthermore, metformin effectively reduced body weight in SLE patients, impacting both patient well-being and cardiovascular risk [4]. A multi-center, randomized, double-blind, placebo-controlled trial (the “Met Lupus” Trial) was conducted to further assess the efficacy and safety of metformin in Chinese SLE patients with low-grade activity (baseline SELENA-SLEDAI ≤6, prednisone ≤20 mg/day) at risk of flares (with a documented flare within 12 months before screening). However, this trial faced recruitment challenges, preventing a definitive conclusion. Post hoc pooled analyses suggested that metformin reduced subsequent disease flares in patients with SLE who had low disease activity, especially in serologically quiescent patients [119], although its impact on plasma autoantibody levels and blood pressure was not determined. It was observed that metformin may have a synergistic effect with hydroxychloroquine, whereas no such effect was observed with other immunosuppressants, such as MMF [119].

Apart from its impact on immune cells, metformin may also mitigate tissue injury associated with lupus manifestations. Oxidative stress and mitochondrial damage have been observed in the kidneys of patients with lupus nephritis, as well as in animal models of the disease [120]. They have also been observed in the liver of lupus-prone mice [121] and in vascular tissues of murine hypertensive SLE mice [17]. Other end-organ targets of lupus pathogenesis likely exhibit similar oxidative stress. It is important to independently evaluate their response to metformin beyond immune cells. Furthermore, SLE patients are more likely to have metabolic syndrome, which includes insulin resistance and elevated fasting blood glucose levels [122,123,124]. Additionally, impaired metabolism predicts coronary artery calcification in women with SLE [125]. Some of these individuals may be prescribed metformin to treat hyperglycemia, potentially lowering the oxidative stress associated with metabolic syndrome and lupus symptoms. However, very few, if any, data have been published to show metformin’s non-immune protective benefits in SLE.

In addition to the beneficial effect of the dual inhibition of glycolysis and mitochondrial metabolism by metformin and 2-DG on autoantibody production and disease progression, other metabolic modulators have also been evaluated in SLE. Rapamycin, which has mTORC1-inhibiting properties, promotes Treg expansion in untouched T cells and reduces disease activity in MRL/lpr mice [126], proving effective in patient’s refractory to conventional treatment [127]. Treatment with N-acetylcysteine, a glutathione precursor that blocks mTOR, reduced mortality in NZBWF1 mice [128] and reduced disease activity in SLE patients [129]. A clinically approved inhibitor of glycosphingolipid biosynthesis, N-butyldeoxynojirimycin, normalized CD4+ T cell functions and decreased anti-dsDNA antibody production by autologous B cells in SLE patients [130]. 3PO, 3-(3-pyridinyl)-1-(4-pyridinyl)-2-propen-1-one, an antagonist of 6-phosphofructo-2-kinase/fructose-2,6-bisphosphatases (PFKFB3), an enzyme controlling a limiting step in glycolysis, prevented the development of imiquimod-induced T cell-mediated delayed-type hypersensitivity and psoriasis in mice [131]. Endothelial cells are highly dependent on glycolysis for migration and proliferation, and PFKFB3 gene deletion in endothelial cells inhibits intraplaque angiogenesis and lesion formation in a murine model of venous bypass grafting [132]. Moreover, the damage to bone marrow endothelial progenitor cells, which are dysfunctional in SLE, was mitigated by PFKFB3 inhibition with 3-PO [133], suggesting potential protective cardiovascular effects of glycolysis inhibition.

It will be important to determine whether lupus mice that develop hypertension exhibit abnormal metabolism in other immune and non-immune cell compartments, such as endothelial cells, and whether the metabolism of these cells is altered following in vivo and in vitro treatment with metabolic modulators.

Despite the relevant data regarding the protective effects of metabolic modulators in reducing autoimmunity and protecting the kidneys in SLE, there is limited evidence regarding their vascular effects. We discovered that the chronic treatment of female NZBWF1 mice with a combination of metformin and 2DG restored splenic Th17/Tregs polarization and reduced disease biomarkers. Remarkably, they improved endothelial dysfunction and prevented the development of hypertension (Appendix A). Likewise, chronic mTORC1 inhibition with rapamycin also enhanced disease activity, vascular oxidative stress, endothelial function, and high blood pressure in female NZBWF1 mice (Appendix A). These vasculo-protective effects were associated with reduced Th17 infiltration. Recently, a randomized double-blind clinical trial study demonstrated that N-acetylcysteine treatment reduced vascular complications in SLE [134]. Idebenone, a synthetic quinone analog of coenzyme Q10, enhances electron transfer chain function by bypassing deficient complex I activity and increasing the amount of ATP synthesized, thereby improving mitochondrial physiology, reducing the aberrant production of mROS, the formation of NETs and the activation of the type I IFN pathway. In fact, in MRL/lpr mice, idebenone-treated mice exhibited a significant reduction in autoimmunity and lupus nephritis, and improved endothelium-dependent vasorelaxation, suggesting that this drug could target SLE vasculopathy [135].

## 5. Conclusions

Numerous pharmacological strategies aimed at restoring dysfunctional metabolism in immune cells, including the inhibition of glycolysis, mitochondrial metabolism, or mTORC1, have shown promising results in improving endothelial dysfunction and preventing the development of hypertension in mouse models of SLE. However, there is limited information available regarding the vasculo-protective effects of drugs targeting immunometabolism in SLE patients. The key points are included in Table 1. 

## Figures and Tables

**Figure 1 biomedicines-11-03142-f001:**
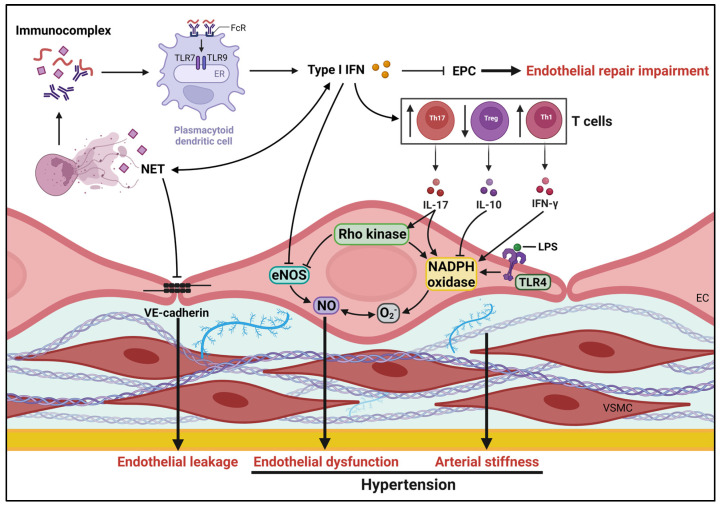
Mechanisms potentially underlying pathogenesis of hypertension in SLE. Several SLE-specific factors, such as impairment of endothelial repair, endothelial dysfunction and arterial stiffness influence vascular health, contributing to development of hypertension. eNOS, endothelial nitric oxide synthase; ECs, endothelial cells; EPCs, endothelial progenitor cells; ER, endoplasmic reticulum; FcRs, Fc receptors; IFN, interferon; IL, interleukin; NETs, neutrophil extracellular traps; NO, nitric oxide; LPS, lipopolysaccharide; TLR, Toll-like receptors; VSMCs, vascular smooth muscle cells.

**Figure 2 biomedicines-11-03142-f002:**
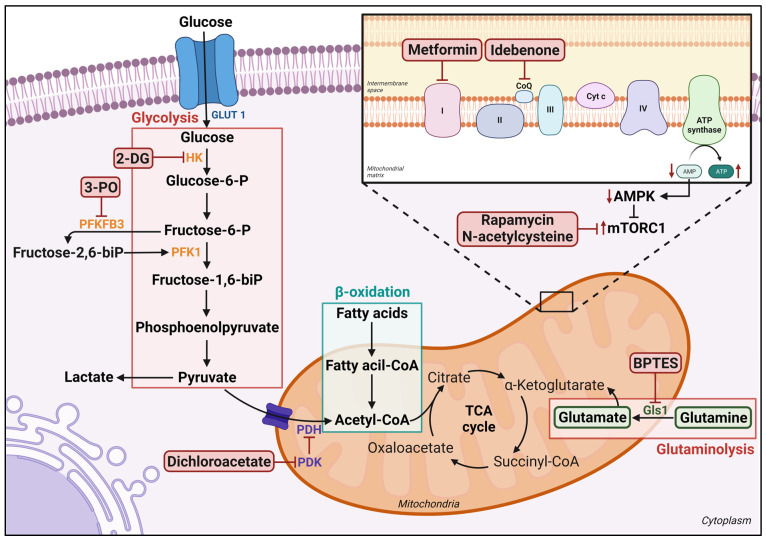
Main metabolic pathway involved on the activation, proliferation, and differentiation of CD4+ T cells, and targets for metabolic modulators. AMPK, AMP-activated protein kinase; BPTES, bis-2-(5-phenylacetamido-1,3,4-thiadiazol-2-yl)ethyl sulfide; CoQ, coenzyme Q10; 2 DG, 2-deoxy-D-glucose; Gls1, Glutaminase 1; HK, hexokinase; PDH, Pyruvate dehydrogenase; PDK, Pyruvate dehydrogenase kinase; PFK1, Phosphofructokinase-1; PFKFB3, 6-phosphofructo-2-kinase/fructose-2,6-bisphosphatase; 3 PO, 3-(3-pyridinyl)-1-(4-pyridinyl)-2-propen-1-one; TCA cycle, tricarboxylic acid cycle; mTORC1, Mammalian target of rapamycin complex 1.

**Table 1 biomedicines-11-03142-t001:** Key points.

Key Points
SLE is associated with several complications, including hypertension and endothelial dysfunction, which promotes a significantly increased risk of stroke and myocardial infarction, although the etiology remains unclear.
The establishment and progression of endothelial dysfunction in SLE are regulated by immunological cells’ imbalance and pro-inflammatory cytokines, together with higher ROS production and arterial stiffness.
The dysregulation of innate and adaptive immune cells in SLE and their infiltration into kidney and vascular tissues are key events associated with cardiovascular complications in SLE, where CD4+ T cells play a crucial role.
T cells depend on glycolysis for inflammatory effector functions (Th1/Th17), while Tregs exhibit higher rates of lipid oxidation, pointing to T cell metabolism as a target for immunotherapy.
Metformin, 2DG, and rapamycin are promising metabolic modulators that have demonstrated beneficial effects on autoimmunity and vascular dysfunction in SLE mice.

## Data Availability

Not applicable.

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
