# Peer review of "Metabolic Modulators in Cardiovascular Complications of Systemic Lupus Erythematosus"

_biomedicines, 2023, doi:10.3390/biomedicines11123142_

Round 1
Reviewer 1 Report
Comments and Suggestions for Authors
This review article summarized the potential favorable outcomes of metabolic modulators in addressing autoimmunity, hypertension, endothelial dysfunction, and renal injury in SLE result from the restoration of immune system equilibrium. The authors proposed that blocking glycolysis, mitochondrial metabolism, or mTORC1 can ameliorate endothelial dysfunction and hinder the onset of hypertension in mouse models of SLE, whereas limited information is known in humans. The topic of this article is interest that provides some insights into the roles and potentials of metabolism-related molecules in lupus-related cardiovascular complications. Before considering accepting this manuscript for publication, there are several issues to consider.
1. The prevalence of traditional cardiovascular risk factors such as hypertension, hypercholesterolemia, diabetes, smoking, and family history in SLE patients will affect metabolic regulators and deserves further discussion.
2. This manuscript is rather extensive. It is advisable to consolidate all summarized essential points into a paragraph or table, facilitating a more reader-friendly format.
3. The authors mentioned some unpublished data in this manuscript. I suggest to include these data as supplementary material to bolster the credibility of the argument. Otherwise, they should cite relevant references.
4. Some descriptions lack appropriate references. For example, L32, “Autoantibodies targeting the cell nucleus are present in 99% of SLE patients, …”; L72-73, “hypertension is main cause for CVD…”; L73-75, “Multiple studies highlight the elevated prevalence of hypertension in women with SLE. Women aged 35 to 44 with SLE are 50 times more likely to experience a cardiac event, such as infarction or angina, than individuals of the same age without the condition [10]”. Is this information in reference 10 the latest?
Author Response
We thank the reviewer for the helpful comments and the positive criticisms. Following his/her suggestions we have made changes to the text, which, we believe, have improved the manuscript.
R#1.1. The prevalence of traditional cardiovascular risk factors such as hypertension, hypercholesterolemia, diabetes, smoking, and family history in SLE patients will affect metabolic regulators and deserves further discussion.
R#1.1. Answer. Following your suggestion, we included more information about the prevalence of traditional cardiovascular risk factors.
R#1.2. This manuscript is rather extensive. It is advisable to consolidate all summarized essential points into a paragraph or table, facilitating a more reader-friendly format.
R#1.2. Answer. We included a table that summarized essential points, as you suggested.
R#1.3. The authors mentioned some unpublished data in this manuscript. I suggest to include these data as supplementary material to bolster the credibility of the argument. Otherwise, they should cite relevant references.
R#1.3. Answer. We included a supplementary figure with unpublished data, as you suggested.
R#1.4. Some descriptions lack appropriate references. For example, L32, “Autoantibodies targeting the cell nucleus are present in 99% of SLE patients, …”; L72-73, “hypertension is main cause for CVD…”; L73-75, “Multiple studies highlight the elevated prevalence of hypertension in women with SLE. Women aged 35 to 44 with SLE are 50 times more likely to experience a cardiac event, such as infarction or angina, than individuals of the same age without the condition [10]”. Is this information in reference 10 the latest?
R#1.4. Answer. Following your suggestion, we included appropriate references.
Reviewer 2 Report
Comments and Suggestions for Authors
This reviewed article is a review type that explores the actual evidence concerning whether the beneficial effects of metabolic modulators on autoimmunity, hypertension, endothelial dysfunction, and renal injury in lupus result from the restoration of a balanced immune system. The subject is very important and actual in the modern rheumatology and the article evaluates several interesting aspects especially in the second part that explores etiopathogenic mechanisms and experimental data on murine models. However, the introductory part requires an update of the existing data and bibliographic references. Also, a wider discussion about the effect of the medication (other than the cyclophosphamide already discussed) would be interesting (there are multiple data on hydroxychloroquine, cortisone but also on the approved biologicals)
Some comments:
Introduction needs some updates and corrections
- Row 30 “persistent autoimmune” disorder – please try to find some other formulation since it is a little bit strange, also “ primarily affects the kidneys” it is not quite true since kidneys are not mandatory the main target in SLE
- Row 34 “even though the presence of anti-dsDNA is predictive in 95% of SLE cases, the exact cause of SLE remains unclear” – you simplified the discussion about the etiopathogenesis of SLE too much, solving it in a single sentence related to dsDNA antibodies
- Rows 29-41 more than just one reference is needed
SLE and hypertension
“Hypertension stands as the primary risk factor for the progression of renal, vascular, and cardiac diseases” please reconsider this phrase- if you are referring to SLE, there are significantly more factors involved in the progression of renal and cardiovascular damage than hypertension. Do not underestimate the immune-mediated etiopathogenesis of these organ damage!
Author Response
We thank the reviewer for the helpful comments and the positive criticisms. Following his/her suggestions we have made changes to the text, which, we believe, have improved the manuscript.
R#2.1. However, the introductory part requires an update of the existing data and bibliographic references. Also, a wider discussion about the effect of the medication (other than the cyclophosphamide already discussed) would be interesting (there are multiple data on hydroxychloroquine, cortisone but also on the approved biologicals)
R#2.1. Answer. Following your suggestion, we have updated the introduction part and have included new bibliographic references. Moreover, we have improved discussion about the effect of medication in cardiovascular complication of SLE.
R#2.2. Row 30 “persistent autoimmune” disorder – please try to find some other formulation since it is a little bit strange, also “primarily affects the kidneys” it is not quite true since kidneys are not mandatory the main target in SLE.
R#2.2. Answer. Following your suggestion, we introduced “Systemic lupus erythematosus (SLE) is a chronic autoimmune disease characterized by multisystemic inflammation and organ damage manifestations that affect the skin, joints, kidneys, heart, lungs, blood and the central nervous system”
R#2.3. Row 34 “even though the presence of anti-dsDNA is predictive in 95% of SLE cases, the exact cause of SLE remains unclear” – you simplified the discussion about the etiopathogenesis of SLE too much, solving it in a single sentence related to dsDNA antibodies
R#2.3. Answer. We rephrased this sentence: “Even though the presence of anti-dsDNA is predictive in 95% of SLE cases, a more extensive immune dysregulation is involved in etiopathogenesis of SLE, although the exact cause of SLE remains unclear”.
R#2.4. Rows 29-41 more than just one reference is needed
R#2.4. Answer. Following your suggestion we include more references.
R#2.5. SLE and hypertension “Hypertension stands as the primary risk factor for the progression of renal, vascular, and cardiac diseases” please reconsider this phrase- if you are referring to SLE, there are significantly more factors involved in the progression of renal and cardiovascular damage than hypertension. Do not underestimate the immune-mediated etiopathogenesis of these organ damage!
R#2.4. Answer. We rephrased this sentence: “Hypertension stands as the leading risk factor for the progression of renal, vascular, and cardiac diseases in the general population [19], which is exacerbated by immune mediated mechanisms in SLE patients.”
Round 2
Reviewer 1 Report
Comments and Suggestions for Authors
All concerns have been appropriately improved.